# Green Conversion of Carbon Dioxide and Sustainable Fuel Synthesis

**Hosam M. Saleh * and Amal I. Hassan**

Radioisotope Department, Nuclear Research Center, Egyptian Atomic Energy Authority, Cairo 11787, Egypt; virtualaml@gmail.com
* Correspondence: hosamsaleh70@yahoo.com or hosam.saleh@eaea.org.eg

**Abstract:** Carbon capture and use may provide motivation for the global problem of mitigating global warming from substantial industrial emitters. Captured $CO_2$ may be transformed into a range of products such as methanol as renewable energy sources. Polymers, cement, and heterogeneous catalysts for varying chemical synthesis are examples of commercial goods. Because some of these components may be converted into power, $CO_2$ is a feedstock and excellent energy transporter. By employing collected $CO_2$ from the atmosphere as the primary hydrocarbon source, a carbon-neutral fuel may be created. The fuel is subsequently burned, and $CO_2$ is released into the atmosphere like a byproduct of the combustion process. There is no net carbon dioxide emitted or withdrawn from the environment during this process, hence the name carbon-neutral fuel. In a world with net-zero $CO_2$ emissions, the anthroposphere will have attained its carbon hold-up capacity in response to a particular global average temperature increase, such as 1.5 °C. As a result, each carbon atom removed from the subsurface (lithosphere) must be returned to it, or it will be expelled into the atmosphere. $CO_2$ removal technologies, such as biofuels with carbon sequestration and direct air capture, will be required to lower the high $CO_2$ concentration in the atmosphere if the Paris Agreement's ambitious climate targets are to be realized. In a carbon-neutral scenario, $CO_2$ consumption with renewable energy is expected to contribute to the displacement of fossil fuels. This article includes a conceptual study and an evaluation of fuel technology that enables a carbon-neutral chemical industry in a net-zero-$CO_2$-emissions environment. These are based on the use of collected $CO_2$ as a feedstock in novel chemical processes, along with "green" hydrogen, or on the use of biomass. It will also shed light on innovative methods of green transformation and getting sustainable, environmentally friendly energy.

**Keywords:** carbon capture; $CO_2$; renewable energy; global warming; methanol; green hydrogen; fossil fuels; biofuels

## 1. Introduction

Global carbon emissions fell 5% in the first quarter of 2020 compared to the first half of 2019, owing to lower coal consumption (8%), oil (4.5%), and natural gases (2.3 percent). Another paper provided the daily, monthly, and annual rhythms of $CO_2$ emissions and anticipated an 8.8 percent reduction in $CO_2$ in the first half of 2020 [1]. The COVID-19 epidemic caused the largest dramatic drop in worldwide $CO_2$ emissions since World War II [2]. The Worldwide Atmospheric Research anticipated that global human fossil $CO_2$ emissions in 2020 will be 5.1 percent lower than in 2019, at 36.0 Gt $CO_2$, barely below the 36.2 Gt $CO_2$ emission level recorded in 2013 [3]. Global carbon emissions (fossil fuels) per unit of Gross Domestic Product (GDP) decreased in 2019, averaging 0.298 t $CO_2$/k USD/year, but per capita carbon emissions were steady at 4.93 t $CO_2$/capita/year, confirming a 15.9 percent increase since 1990 [4].

To minimize the magnitude of global warming, substantial reductions in $CO_2$ emissions from fossil fuel use are necessary. In view of the steadily increasing interest in

preserving the environment and mitigating the consequences of climate change and global warming, hence, great attention is dedicated to finding alternative sustainable methods or eco-friendly economic materials to reduce the use of cement in various fields and consequently reduce the $CO_2$ emissions accompanying with intensive energy consumed in the cement industry. Various additives have been mixed with cement such as bitumen [5,6], asphaltene [7], glass [8,9], polymers [10–12], nanomaterials [13,14], cement wastes [15,16], bio-waste [17–19] or natural clay [20,21] to improve the material features and minimize the cement incorporation and environmental pollution.

However, carbon-based liquid fuels will continue to be key energy storage media for the foreseeable future [22]. We suggest using solar energy to recycle atmospheric $CO_2$ into liquid gasoline using a mix of mostly available technologies. The world population hit 7 billion in 2011 and is expected to exceed 9 billion by 2050, according to the United Nations, significantly increasing energy consumption. According to Hubbert's peak theory, oil resources will plummet dramatically during the next 40 years [23]. As a result, the years 2055–2060 may be the last in which oil output reaches zero. Furthermore, while taking current energy usage into account, the World Coal Institute estimates that current coal reserves will last another 130 years, oil reserves for 42 years, and natural gas reserves for 60 years. The need to replace old fossil fuels with sustainable alternatives and new energy scenarios is thus becoming critical. Another major environmental worry is the rising quantities of carbon dioxide ($CO_2$) in the atmosphere, which is caused by the combustion of fossil fuels and contributes to global warming [24].

Carbon capture allows for the production of low-carbon power from fossil fuels as well as the reduction of $CO_2$ emissions from industrial operations, such as gas processing, cement production, and steel production, where alternative decarbonization options are restricted. $CO_2$ capture and usage is gaining popularity across the world [24].

Human-caused $CO_2$ increases have disastrous repercussions, such as rising average temperatures, melting sea ice, flooding, and sea level rise, harming global life, health, and the economy. The staggering difference between the total amount of carbon emitted worldwide as $CO_2$ and the quantity of carbon present in five crucial carbon-containing commercial chemicals, not including derivative products such as ethylene oxide [25], underscores the grave consequences of continued carbon emissions. Even with indirect effects, the greatest quantity of $CO_2$ that may be used remains quite little in comparison to total emissions. For example, if the whole worldwide yearly manufacturing of ethylene, the most extensively made commercial chemical containing carbon atoms, were carried out solely using carbon derived from $CO_2$, this would result in direct usage of less than 1.5 percent of total global $CO_2$ emissions [26]. Even with additional commercial chemicals included and indirect mitigation assumptions applied, possible usage might constitute just a tiny fraction of overall emissions. Naturally, this ignores the reality that direct $CO_2$ conversion into hydrocarbons, particularly aromatics, is unlikely to become a commercial method in the near future.

Nonetheless, $CO_2$ activation is exceedingly difficult since $CO_2$ is a completely oxidized, thermodynamically stable, and chemically inactive molecule. Furthermore, hydrocarbon synthesis by $CO_2$ hydrogenation frequently results in the creation of undesirable short-chain hydrocarbons rather than the desired long-chain hydrocarbons [27]. As a result, most research in this field has been on the sequential hydrogenation of $CO_2$ to $CH_4$, oxygenates, $CH_3OH$, $HCOOH$, and light olefins ($C_2$–$C_4$ olefins). There have been few experiments on creating liquid hydrocarbons with molecular weights of $C^{5+}$.

There are two methods of converting $CO_2$ to hydrocarbon gas liquids: an indirect pathway that transforms $CO_2$ to carbon monoxide (CO) or methanol and then to liquid hydrocarbons, or a direct $CO_2$ hydrogenation route that combines $CO_2$ reduction to CO through the use of the converse water-gas shift reaction shift (CWGS) and subsequent CO hydrogenation to long-chain hydrocarbons through the use of the Fischer-Tropsch synthesis (FTS) [28,29]. Secondly, the more direct technique is typically viewed as cheaper and ecologically acceptable since it requires fewer chemical process stages and uses less energy

overall. The following chemical reactions are relevant for the production of hydrocarbon fuel (Scheme 1).

**Hydrogenation of $CO_2$:**

$CO_2 + 3H_2 \rightleftharpoons CH_2 + 2H_2O$ ($\Delta H^0_{298} = 125$ kJ mol$^{-1}$;

The CWGS reaction: $CO_2 + H_2 \rightleftharpoons CO + H_2O$ ($\Delta H^0_{298} = 41$ kJ mol$^{-1}$); and

The FTS reaction: $CO + 2H_2 \rightleftharpoons -(CH_2)- +H_2O$ ($\Delta H^0_{298} = -166$ kJ mol$^{-1}$ )

**Scheme 1.** Chemical reactions used to produce hydrocarbon fuels.

Mitigation of greenhouse gases is one of the most pressing issues confronting society today. As a result, the best strategy to minimize greenhouse gas emissions is to employ carbon-free sources that do not emit more $CO_2$ into the environment [30]. However, there is significant promise in $CO_2$ energy carriers and other materials, with several hurdles to overcome. Reduced $CO_2$ emissions and conversion to renewable fuels and valuable chemicals have been proposed as possible ways to reduce greenhouse gas emissions [30].

This review discusses the most current breakthroughs and problems in $CO_2$ use for effective renewable fuel (RF) synthesis. This comprises revolutionary technologies, techniques, and existing impediments to $CO_2$ to RF conversion, with the goal of establishing advancements in this field and presenting an overview of recent research trends for the prospective development of new concepts for large-scale $CO_2$ reduction into methanol.

## 2. Global Warming and Sustainability

Producing energy from $CO_2$ rather than absorbing and disposing of it is a sustainable energy technique. In reality, several thermochemical processes may transform $CO_2$ into value-added compounds such as hydrogen, methanol, and ethanol [31]. These fuels may be utilized to power automobiles without requiring major modifications to existing internal combustion engines, paving the way for a more sustainable future. Discovering eco-friendly alternatives and new energy scenarios, primarily the conversion of $CO_2$ into value-added chemicals using sunlight, is one of the top-most research priorities to overcome today's extreme reducing the use of fossil fuels and mitigate the global warming risks associated with high $CO_2$ concentrations in the atmosphere [32].

Because of decreased prices, quick commercial adoption, and complete industrial development, the use of second-generation green fuels is one of the most promising approaches to achieving industrial and ecological aims in the short term. A green fuel, unlike fossil fuel, does not emit more $CO_2$ into the atmosphere since its burning, in the well-to-wheel cycle, creates a $CO_2$ release equivalent to that required for its manufacture, closing the $CO_2$ balance, and making it a carbon-neutral fuel [33]. The practical application of hydrogenation catalysts in conjunction with commercially viable research and innovation technology. Indeed, catalytic conversion of atmospheric $CO_2$ into biofuels and fine chemicals would be one of the most economical and realistic solutions to the problem of greenhouse gas emissions, assuming that capture/sequestration and high-pressure storage technologies become commercially viable [34,35]. Nonetheless, in a very short time, the most prosecutable actions and encouraging prospects look upon the use of $CO_2$-rich streams originating from industrial exhaust emissions, such as brick and cement work, despite the need for clean-up stages, as well as additional purification and concentration [36]. Haldor-Topsoe is at the vanguard of using $CO_2$ as a carbon feedstock through various power-to-gas technologies, whereas ENI is redesigning its production lines with a greener vision, by developing new hydrogenation processes, called ENI Ecofining$^{TM}$, for green-fuels synthesis, advantageously depicting the production of ultra-pure $CO_2$ as industrial

practice [37]. Despite significant advances in fuel science, the commercial and economic viability of hydrogen-to-liquid-fuels such as methanol is still being debated. The discovery of more effective catalyst substances for the efficient hydrogenation of $CO_2$ looks to be of major importance in this scenario. CuO/ZnO-based catalysts, as is well known, offer lower costs and improved chemical stability when compared to other catalysts such as transition metal carbides (TMCs), bimetallic catalysts, or Au-supported catalysts [38]. Although $CuO/ZnO/Al_2O_3$ is the most extensively investigated catalyst for methanol synthesis, it was shown that the inclusion of $ZrO_2$, $CeO_2$, and $TiO_2$ oxides improved both the activity and selectivity of CuO/ZnO-based catalysts in $CO_2$ hydrogenation processes [39].

Recently the influence of replacing $Al_2O_3$ with $CeO_2$ in the typical $Cu-ZnO/Al_2O_3$ catalytic discussed composition for syngas conversion, suggesting that the use of cerium oxide led to a remarkable positive effect on the catalyst stability, as a result of the preservation of the ratio of $CuO/Cu^+$ on the catalyst surface. As well as, on the particle growth, due to the strong electron interactions of the copper/ceria phase ($Cu/Cu_2O/CuO$, $Ce^{3+}/Ce^{4+}$) during hydrogenation reactions [40]. Metal dispersion and $CO_2$ adsorption capacity are two key parameters affecting the $CO_2$ hydrogenation functionality of Cu-based systems. Similarly, in our previous works, we displayed that the use of several oxides (i.e., ZnO, $ZrO_2$, $CeO_2$, $Al_2O_3$, $Gd_2O_3$, $Ce_2O_3$, and $Y_2O_3$) and alkaline metals (i.e., Li, Cs, and K) could remarkably influence both catalyst structure and morphology, balancing the amount of the diverse copper species (i.e., $Cu^{\circ}/Cu^+/Cu^{2+}$) and leading to a notable improvement of the catalytic performance. On this account, $ZrO_2$ has been shown to positively the affect morphology and texture of Cu-ZnO based catalysts, also favoring $CO_2$ adsorption / activation and methanol selectivity, while $CeO_2$ could act as both an electronic promoter and improver of surface functionality of Cu phase [41]. As reported in our preliminary works, the scientists have proved a greater specific activity of $CuZnO-CeO_2$ catalyst with respect to that of similar catalytic systems containing $ZrO_2$ or other promoter oxides, facing several electronic and structural effects. Despite of a generally higher specific activity, we have evidenced that the CuZnO-Ceria catalyst suffers from several practical limitations, such as: a lower surface area, especially compared to that of $ZrO_2$ and $Al_2O_3$ promoted catalysts.

## 3. Green Hydrogen and Biofuel

Hydrogen has several benefits as an energy carrier and will play a key role in future energy production and circular economy [42]. Because hydrogen can be created in a climate-neutral way and emits only water when used, it is regarded as a climate-friendly energy carrier [43]. Hydrogen derived from renewable sources is commonly referred to as "green hydrogen." It contrasts with "grey hydrogen," which is defined as hydrogen derived from fossil, nonrenewable sources [44]. Currently, grey hydrogen generated by steam reforming of fossil natural gas accounts for the majority of worldwide hydrogen production [44]. Grey hydrogen may be converted to "blue hydrogen" by using carbon capture and storage technology, but it remains a non-renewable source [45]. To distinguish between carbon-neutral green and blue hydrogen, we coined the term "carbon-negative hydrogen." Hydrogen has a wide range of uses due to its high gravimetric energy density, as well as its ease of storage and transportation [46]. Fuel cells may produce both electric power and heat from hydrogen. In the steel industry, hydrogen may also be utilized as a reducing agent [46]. It is regarded as an unavoidable basic building element in difficult-to-electrify zones [46]. Green hydrogen may be created using renewable power or renewable biomass, either through thermochemical processes or using microorganisms and biotechnological technologies. Biohydrogen, or hydrogen generated from biomass, can lead to more effective use of biogenic source materials by using residual and waste materials. Biohydrogen production technologies are classified as thermochemical or biotechnological. Biohydrogen may be generated on an industrial scale utilizing thermochemical methods from biomass (wood, straw, grass clippings, etc.) as well as other bioenergy sources (biogas, bioethanol, etc.) [43]. Gasification and pyrolysis are two types of thermochemical reactions. The synthesis gas produced by gasification or pyrolysis contains varying proportions of

$H_2$, $CO_2$, carbon monoxide (CO), methane ($CH_4$), and other components such as nitrogen, water vapor, light hydrocarbons, hydrochloric acid, alkali chlorides, Sulphur compounds, biochar, or tar [47]. The following chemical processes occur as a result of subsequent steam reforming and/or water gas shift reactions, culminating in the creation of $CO_2$ and hydrogen [48]. Minimal carbon-hydrogen (MCH) should be used initially in areas that are difficult to electrify directly [49]. We believe that generated bio-hydrogen is employed in difficult-to-electrify industrial sectors such as cement, steel, refining, ammonia, and glass. Refineries and ammonia plants are now the largest customers of hydrogen generated by steam methane reforming, a process that releases at least 10 kg $CO_2$ for every kilogram $H_2$ produced. Because many industries utilize carbon-intensive hydrogen, switching to MCH would be a near-term opportunity to increase hydrogen demand while lowering greenhouse gas emissions. Companies that are difficult to electrify may reduce their emissions and capture carbon dioxide by switching to alternative fuels and implementing carbon capture and storage technologies. This can help to enable their operations to contribute to the elimination of carbon dioxide from the atmosphere.

Country-specific forecasts of European nations' ultimate energy consumption in 2050 are unavailable and difficult to determine. As a result, scientists evaluate country-specific final energy consumption in 2019 and anticipate that 5–30% of final energy consumption will be fulfilled with low-carbon hydrogen to minimize hard-to-electrify emissions. These scenarios enable us to set upper and lower bounds on the amount of hydrogen likely required by each European country to achieve net-zero emissions by 2050 [50].

## 4. Carbon Dioxide Sequestration by Microalgae

Carbon dioxide ($CO_2$) sequestration and stabilization technologies are methods designed to reduce the amount of carbon dioxide emissions in the atmosphere, in order to mitigate the effects of climate change. $CO_2$ sequestration refers to the process of capturing and storing $CO_2$ from industrial processes or power generation, while $CO_2$ stabilization refers to methods that aim to reduce the amount of $CO_2$ emitted into the atmosphere. There are several types of $CO_2$ sequestration technologies. Carbon capture and storage (CCS) is a proven technology that has been used for several decades in various industries. It has the potential to capture a significant portion of $CO_2$ emissions from industrial processes and power generation, and to store them safely underground [51]. However, CCS is currently expensive and requires significant infrastructure, and there are concerns about the long-term integrity of storage sites and potential leaks [51]. Direct air capture (DAC) is an emerging technology that has the potential to capture $CO_2$ directly from ambient air, regardless of the source of emissions [52]. It could be useful for decarbonizing industries that are difficult to electrify, such as aviation or shipping. However, DAC is currently expensive and energy-intensive, and there are concerns about the scalability of the technology [52]. Carbon capture and utilization (CCU) can help to reduce emissions while also creating new economic opportunities [53]. It has the potential to be cost-effective, especially if the captured $CO_2$ can be used to produce high-value products. However, the scale of CCU is currently limited, and it may not be feasible for all industries or applications.

Enhanced oil recovery can help to reduce emissions while also increasing oil recovery. It has been used for several decades and is a well-established method. However, it is limited to areas with suitable oil reservoirs, and the potential for $CO_2$ leakage or other environmental risks must be carefully monitored. Mineral carbonation is a promising method that has the potential to store $CO_2$ in a stable and long-lasting form [54]. It could be particularly useful for capturing $CO_2$ from industrial processes that produce high-purity $CO_2$ streams. However, the process is currently expensive and requires significant energy inputs.

It is critical to improve existing technology and find appropriate replacements. In comparison to physical and chemical approaches, biological $CO_2$ fixation (CDF) appears to be a more cost-effective and environmentally beneficial solution. Photosynthetic organisms ingest $CO_2$ during the dark period of photosynthesis and serve an important role in

maintaining the equilibrium of $CO_2$ levels in the atmosphere [55]. Phytoplankton showed better CDF capacity and biomass output when compared to other green organisms [56]. Marine phytoplankton provides for half of the total global primary productivity, fixing 50 gigatonnes of $CO_2$ every year [57]. In this context, experiments on carbon capture by microalgae have piqued the interest of scientists all around the world. Microalgae can absorb $CO_2$ 10–50 times more efficiently than vascular plants while not competing with or supplying food for humans/animals [57]. The carbon concentration process is a specific mechanism used by microalgae to ingest carbon dioxide. The pyrenoid, a specific organelle, raises the level of $CO_2$ near the thylakoid membranes in this process [58]. The rising carbon dioxide content near the thylakoid membrane increases the effectiveness of ribulose-1,5-bisphosphate carboxylase/oxygenase, a key photosynthetic enzyme for carbon uptake or sequestration.

In order to accomplish sustainability goals, not only economic elements but also social (protection or hazard-free), environmental (pollution prevention and regulatory authority), energy, material, and economic performance aspects of processes and products must be reviewed, optimized, and regulated [59]. Furthermore, these process control systems should protect against unforeseen instabilities that occur when the process deteriorates over time and/or because of changes in feed composition flow rate, temperature, and pressure. Furthermore, a hierarchical design technique was presented to synthesize economically efficient separation procedures while considering environmental concerns as limitations [60]. Recently, a modular strategy for designing sustainable chemical processes was established by combining quantitative economic and environmental variables with qualitative social indicators [61]. Multi-objective optimization systems have a greater chance of achieving the best trade-off between competing economic and environmental goals. For instance, a universal optimization technique for sustainability was created, in which a large-scale algae processing network was concurrently optimized in terms of lowering the unit cost- and global warming ramifications indicators [62]. A multi-objective genetic algorithm was employed to address an integer non-linear programming problem related to environmental concerns, despite it having a single objective. The algorithm was able to optimize for multiple factors simultaneously, allowing for a more comprehensive and effective solution. Sustainable process control research for chemical processes, on the other hand, is not as well recognized as sustainability design optimization. There has been little published research on process operations that use sustainability-oriented control techniques [63]. To address the sustainability of a batch reactor, a strategy incorporating stochastic dynamic optimization with optimum control was presented [64]. Another use of deterministic optimum control techniques to increase energy efficiency in industrial processes has been documented [65]. Only utility-related environmental consequences were examined in these two investigations. Given the contradicting nature of sustainable indicators, this constraint might be linked to a lack of effective solutions for integrating process sustainability features into the advanced controller architecture (e.g., economics vs. environment) [65]. According to the principles of green chemistry and engineering [66], chemical production processes that reduce or limit greenhouse gas emissions of hazardous substances should be developed by minimizing waste, performing real-time analysis regulating contamination and dangerous incident prevention, maximizing mass, energy, space, and time efficiency, and so on [66]. Furthermore, when improved sustainability is established, it must be maintained in the face of any beyond-the-gate (front-edge alterations) and/or process stage disturbances. Catalytic, electrochemical, mineralization, biological (using microorganisms and enzymes), photocatalytic, and photosynthetic processes are among the various technical pathways available for converting $CO_2$ into commercial goods [67]. Electrolyzers are used in electrochemical processes to convert $CO_2$ to CO. $H_2$, which is often produced by water electrolysis, is a common co-reactant in the conversion of $CO_2$ to $CH_4$, $CH_3OH$, and other compounds. Electrolysis techniques are energy-intensive and expensive [68]. Near-zero emission energy sources must be employed for the systems to be carbon-neutral. $CO_2$ electrolysis is a more new study topic than $H_2O$ electrolysis, which is well known, and thus

more effort is needed to develop affordable, robust catalytic materials with high efficiency, selectivity, and yield [26]. Energy efficiency, Faradaic efficiency, conversion rate, long-term catalyst(s) stability and durability, and process economics are five critical system-level aspects to consider when commercializing these technologies. The primary distinction between photocatalytic and electrochemical $CO_2$ reduction is the source of electrons, which are acquired in the former by exposing semiconductors to light and in the latter by providing a current [68]. The direct utilization of photons, as opposed to the first conversion into electricity, is a significant benefit of the photocatalytic process. However, such processes are complicated, including several mechanisms that are not fully understood, such as electron and proton transport and chemical bond formation/breaking [68]. Solar-energy-driven $CO_2$ conversion has sparked tremendous interest across the world, with research focusing mostly on the creation of innovative nanomaterials photocatalytic materials and laboratory-scale examination of the reaction process. Efficiency is critical for affordability and scalability in any system that uses solar power for $CO_2$ conversion [69]. Currently, the attainable $CO_2$ conversion rates of photocatalytic/photothermic catalytic systems in development are frequently low and impractical for commercial-scale operations. Because the electrochemical or photocatalytic reduction of $CO_2$ to create high-specificity chemicals or fuels is difficult, it may be easier to employ some well-established catalytic techniques to react $CO_2$ and $H_2$ to form carbon-based products [70]. The catalyst—the substance that transforms $CO_2$ that must have high efficiency, selectivity, quick reaction rates, and stability—is at the core of various $CO_2$ conversion methods. $CO_2$ conversion catalysts are actively researched and developed across the world [71]. A number of catalysts have been developed and tested, and they have proven to be capable of converting $CO_2$ into a variety of compounds with high efficiency, selectivity, and yields. Several papers available discuss current advances in catalyst reactor design [72]. More research is needed to make these catalysts more efficient, selective, and stable over time. More effort is needed to create technologically and economically feasible procedures for the commercial conversion of $CO_2$ into fuels and chemicals based on these catalysts. Furthermore, one crucial technology for the competitiveness of $CO_2$-derived chemicals and fuels on the market is the low-cost synthesis of carbon-free H2 [73]. These reactions need pure $CO_2$; hence, $CO_2$ released from sources such as fossil fuels are insufficient. In general, these processes demand pure $CO_2$, hence $CO_2$ released from sources such as fossil-fuel power plants, steel, and cement production must be cleaned, raising the prices even higher. Biotechnology relies on the capacity of living organisms such as microbes, algae, or plant cells to metabolize one material into another.

Numerous studies over the last decade have demonstrated that $CO_2$ mitigation using algae is a sustainable process with the simultaneous synthesis of high-calorific products such as biodiesel, pigments, fatty acids, and so on [74,75]. The broad dispersion, high biomass output, capacity to respond under harsh conditions, rapid carbon absorption and consumption, and ability to develop value-added products all contribute to algae's potential usage (Figure 1 and Table 1). The carbonic anhydrase enzyme is used by both macro and microalgae to digest inorganic carbon via a photoautotrophic process [76]. The $NADH_2$ produced by the electron transport chain joins with the Ribulose-1,5-bisphosphate carboxylase/oxygenase (supplied by the carbonic anhydrase enzyme) and aids in carbohydrate creation from $CO_2$ as well as providing limited power in the Calvin cycle for glucose synthesis [77]. There were also numerous more strains of microalgae that grew rapidly under harsh environments such as high pH and temperature [78]. Lipids, which may be utilized as a source for biofuel production (e.g., biodiesel), are the most industrially useful algal product [79]. As a result, a diverse set of strains for high lipid synthesis and boosting growth parameters like as light, temperature, and pH are critical to increasing productivity [80]. Unoptimized pH during algae development can influence carbon species distribution and carbon availability. Furthermore, at high pH levels, it can have a direct impact on the metabolic activity of algae.

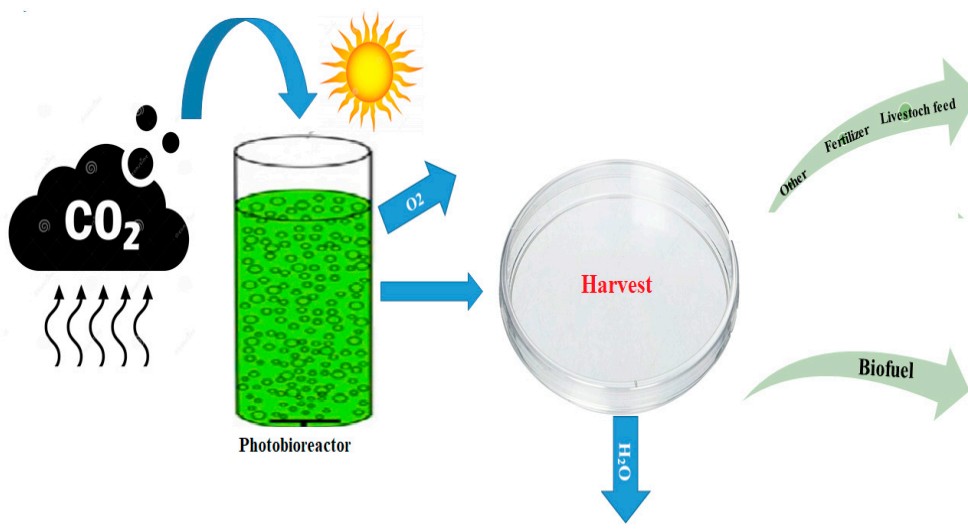

**Figure 1.** Green conversion of $CO_2$ to biofuel.

Green algae species such as *Chlorella vulgaris* and *Nannochloropsis gaditana* were studied for CDF in tubular bioreactors under optimal conditions of temperature (25 C), $CO_2$ concentration (4 and 8 volume percent), and artificial light [81]. The greater CDF in both species was sustained by an 8 volume percent $CO_2$ concentration. When compared to other species, *Nannochloropsis gaditana* (*N. gaditana*) had a greater $CO_2$ biofixation rate (1.7 g/L/day) and cell concentration. Overall the biological control applications offer a promising solution for reducing carbon dioxide emissions and bioconverting them into renewable fuels (Table 1). Researchers have been exploring the use of enzymes found in nature to catalyze the conversion of carbon dioxide into sustainable fuel. Enzymes are highly efficient and selective catalysts, and by mimicking their structure and function, researchers hope to develop new catalysts with improved performance and sustainability.

**Table 1.** Some biological control applications for carbon dioxide stabilization and bioconversion into renewable fuels.

| Target | Method | Enzymes | Species | Product | Pathway | References |
|---|---|---|---|---|---|---|
| Changing heterotrophic yeast into autotrophs capable of utilizing carbon dioxide as their sole carbon source | Isotopic tracing, metabolic rewiring, gene modification, responsive laboratory evolution | cbbM, groEL, groES, PGK1, TDH3, TPI1, TKL1, | *Pichia pastoris* | Biomass | Calvin-Benson-Bassham (CBB) | [82] |
| *C. carboxidivorans* homologous and heterozygous gene upregulation to improve ethyl alcohol and butanol manufacturing | Expression of homozygous and heterozygous RT-PCR, fermentation, and enzyme methods | aor, adhE2, fnr | *Clostridium carboxidivorans* | Butanol, ethyl alcohol | Wood-Ljungdahl pathway, alcohol biosynthesis | [83] |
| To induce artificial autotrophy in Escherichia coli. | Isotopic tracing, metabolic rewiring, gene modification, responsive laboratory evolution | Rubisco, Prk, CA, FDH, | *Escherichia coli* | Biomass | CBB | [84] |

| Target | Method | Enzymes | Species | Product | Pathway | References |
|---|---|---|---|---|---|---|
| To design an efficient ATP path that leads to fixing of carbon dioxide and malate biosynthesis. | UPLC-MS/MS UPLCMS HPLC Genetic manipulations Gene overexpression, Enzyme assays, gas chromatograph, | ΔpfkAB, Δzwf | *S. elongatus Escherichia coli* | Malate | PCK CBB | [85] |
| To cultivate artificial autotrophs in *M. extorquens*. | Growth phenotyping, whole-cell proteomics LC-MS/MS, Spectrophotometric assays, genetic manipulation | PCK | *Methylobacterium extorquens* | Biomass | CBB | [86] |
| To direct carbon flow toward the shikimate pathway in order to generate aromatic compounds from $CO_2$ sequestration. | Metabolic engineering, flow rate analysis, gene modification | MDH | *S. elongatus* | Aromatic compound: 2-phenylethanol, phenylpyruvate, L-phenylalanine, phenylacetalde-hyde | Shikimate | [87] |
| To build a heterogeneous route in Anabaena sp. for 1,3-propanediol syntheses. | Homologous recombination, pathway optimization, qPCR, GC-MS, and HPLC are all examples of gene modification. | Rubisco | *Anabaena PCC7120* | 1,3-Propanediol | 1,3-propanediol biosynthetic | [88] |
| To create an effective hybrid (bioelectrochemical) structure for carbon dioxide fixation. | CRISPR-Cas9 gene output, gas chromatography, mass spectrometry, and electrochemical cultivation, gene cloning. | Prk | *Escherichia coli* | Formate | RGP, rGCS | [89] |
| Using engineered bacteria, alkanes are produced from carbon dioxide. | Gene cloning, biosensor, biofluorescence, LC | Acr1, Ramo, Aar, seado, pmado, cer1 | *Acetobacterium woodii, Acinetobacter baylyi* | Alkanes | Alkane biosynthesis | [90] |

Two microalgae species (*Chlorella sp.* and *Tetraselmis suecica*) were studied for growth kinetics, pH, and biofixation at various $CO_2$ concentrations (0, 5, 15, and 30%) [91]. Both species demonstrated varying degrees of tolerance to $CO_2$ concentrations. The optimal pH varied from 7.5 to 9. Both species' biomass were fermented by Clostridium saccharoper-butylacenaticum N1 4, which generated value-added biochemicals such as organic acid (acetic acid and butyric acid) and solvent (acetone, butanol, and ethanol). *Chlorella vulgaris* was tested using various colored light filters and unfiltered light [92]. The white-colored filter had the highest biomass content (2.26 g/L). The important factors of the process, such as carbon sources and concentrations, pH conditions, buffering agent, $H_2$ gas (as electron donor), and $CO_2$ availability, were investigated in the manufacturing of succinic acid using a strain strongly tied to *Citrobacter amalonaticus* [93]. The study's findings revealed that utilizing sucrose as a substrate including $H_2$ and $CO_2$ resulted in the highest synthesis of succinic acid (12.07 g/L), as well as the removal of by-products (formic, lactic, and acetic acids). A 900 L tangential spiral-flow column photo-bioreactor was assessed in a recent study [94] to optimize the light distribution and light/dark cycle for better $CO_2$ fixation utilizing *Arthrospira sp.* The higher cell concentration aided total photosynthesis and improved $CO_2$ fixation by 59%. The cells obtained maximal values of chlorophyll a content (8.769 mg/L), helix pith (78.26 m), and $CO_2$ fixation rate of 0.358 g/L/day with a

$CO_2$ aeration rate of 0.210 L/min, the $CO_2$ volume concentration of 15%, and circulating pump power of 30 W. For kinetic modeling of *Chlorella vulgaris* growth and $CO_2$ fixation, a central composite design-based statistical technique was used [95]. Temperature, $CO_2$ concentration, nutrients (N, C, and P), gas flow rate, light intensity, and initial inoculum concentration were all taken into account. Under optimized conditions, the highest optimum growth and $CO_2$ fixation with an energy ratio of 19.5 were obtained: temperature, 25 °C; $CO_2$ concentration, 20%; gas flow rate, 0.5 vvm; total nitrogen, 19 mg-N/L; total phosphorous, 7 mg-P/L; initial concentration of inoculum, 0.52 mg/L; and light intensity, 150 $\mu E/m^2 s$. Approximately one-third of the sugar substrate is lost as $CO_2$ during microbial fermentation (pyruvate decarboxylation to acetyl-CoA) [96]. *Clostridium acetobutylicum* (which eats sugars to create a range of compounds in addition to releasing $CO_2$ and $H_2$) and *Clostridium ljungdahlii* (which fixes $CO_2$ via the Wood-Ljungdahl pathway) were employed to repair the released $CO_2$ [97]. This co-culture was successful in recovering carbon and producing non-native metabolites isopropanol and 2,3-butanediol. Based on the $H_2$ substrate, the wood-Ljungdahl route is efficient for creating acetate and ethanol. This method has been used to successfully produce alkanes [98], acetone and acetate, and lipids for biodiesel synthesis from $CO_2$.

Microalgae have a high biomass production and are resistant to harsh environmental conditions. As a result, microalgae are being explored as a possible feedstock for Sequestration and bioenergy generation [78,99]

The growing methodology, microalgal strain, flue gas composition, $CO_2$ tolerance capability, and other factors all contribute to the issues involved with algae-based carbon sequestration [78]. In the biomass cultivation stage, operating costs and energy consumption are increased. Several studies have shown that an open lagoon is more expensive for algae production, but it is not a suitable choice for preserving cultural integrity. The dynamics of heat transmission, irradiance, and nutrient status have been explored, and it has been discovered that thermal modeling is critical for the open raceway, as outlined in [100]. Life cycle assessment (LCA) is an effective method for measuring input-output inventories, power consumption, cost, and the entire life cycle. The LCA of microalgae carbon sequestration begins with microalgae growing and progresses to product creation. A detailed LCA on microalgae generation and flue gas sequestration in raceways ponds revealed that the semicontinuous culture system had a 3.5-fold greater rate of growth for biomass productivity than batch cultivation. The study also found that flue-gas-fed outdoor raceway ponds (RPs) might reduce GHG emissions by 45–50% compared to the baseline [100]. According to another LCA assessment, the cost of producing biomass is 4.87 USD per kilogram, and the energy usage is 0.96 kWh/kg of *Chlorella* biomass. Under natural daylight circumstances, 4000 $m^3$ algae culture ponds might sequester up to 2.2 k tonnes of $CO_2$ per year, according to one research [101]. Another research found that 50 MW power stations may emit 414,000 t/year $CO_2$, whereas a 1000-ha open RPs could absorb 250,000 t/year $CO_2$. According to this study, algae might cut $CO_2$ emissions by 50%. A primary obstacles to indirect $CO_2$ extraction from industrialized flue gas utilizing microalgae is that flue gas includes 142 chemical components that are potentially harmful to microalgae [4]. The low amounts of NOx and SOx in flue gas can provide nutrients to microalgae, while greater concentrations can be hazardous. $CO_2$ solubilization (CDs) in culture media is primarily determined by pH, temperature, and salt content. Numerous growth media have high salinity, which raises osmotic pressure and, as a result, inhibits CDs in the medium. Feed-batch cultures, for example, can be used to progressively deliver the salts required for microalgal development [102]. Various studies have demonstrated the specialized suitability of algal $CO_2$ sequestration; however, the crucial problems are the key and all circumstances of improvements that will enhance the financial functionality of algal CDS and contribute to making this a rational advanced method to deal with GHG remediation. Microalgae carbon sequestration is a long-term strategy for reducing global $CO_2$ emissions [103]. Microalgae is a more appropriate alternative for carbon emission reduction due to recent advancements in culture conditions, harvesting, CDS capability,

and LCA studies [103]. The majority of research has been on determining how to select and cultivate several potential microalgae species for effective CDS. There has been little focus on the establishment of large and commercial-scale carbon sequestration utilizing microalgae. Most of the study has focused on how to identify and develop several possible microalgae species for successful CDS. There has been a minimal emphasis on the development of large-scale and commercial-scale carbon sequestration using microalgae. Though research in algal carbon sequestration mostly concerns strain mutation, improved growth techniques, and harvesting, there is a vacuum in the area involving variables influencing carbon sequestration (e.g., change in light intensity over the day (month, and year), and so on. Likewise, pyrenoid is the most significant sub-organ of microalgae that plays a vital function (as stated above in a separate area) in the carbon concentration process, but research in this field is still in its early stages. Furthermore, research on high-$CO_2$-tolerant microalgae is required to improve carbon sequestration. More research is needed to determine how to minimize the loss of residual undamaged carbon [104].

However, there are also some challenges to overcome in using biotechnology for $CO_2$ sequestration. For example, the productivity and stability of the organisms used must be optimized to ensure that they can capture and convert $CO_2$ efficiently and consistently. Additionally, the processing methods for the products produced must be cost-effective and scalable to enable commercial viability [105]. The carbon neutrality of green fuels is achieved through the use of renewable resources such as biomass, wind, or solar energy to produce the fuel. During the production process, renewable resources absorb carbon from the atmosphere through photosynthesis or other natural processes [106]. This carbon is converted into fuel through various conversion processes, such as fermentation or pyrolysis. When green fuel is burned, it releases carbon emissions into the atmosphere, but these emissions are balanced by the carbon absorbed during the production process. This creates a closed carbon cycle, where the carbon emitted during fuel use is reabsorbed during the production process, resulting in no net increase in carbon emissions. It is important to note that the carbon neutrality of green fuels depends on various factors, including the source of raw materials, the efficiency of the conversion process, and transportation emissions. A comprehensive lifecycle analysis is necessary to accurately assess the carbon footprint of green fuel and determine whether it is carbon-neutral [107].

Overall, biotechnology, particularly the use of microbes, algae, or plant cells to metabolize one material into another, is a promising approach for $CO_2$ sequestration and stabilization. Further research and development will be needed to optimize the efficiency, selectivity, and cost-effectiveness of these approaches, but they hold significant potential for addressing the challenges of climate change and reducing greenhouse gas emissions.

## 5. Future Perspective

The green conversion of $CO_2$ into sustainable fuels is a rapidly evolving field, with many promising developments on the horizon [69]. The integration of multiple technologies and approaches to optimize the $CO_2$ conversion process. For example, researchers are exploring the use of renewable energy sources, such as solar and wind power, to power the $CO_2$ conversion process. This approach would enable the production of sustainable fuels without relying on fossil fuels, further reducing greenhouse gas emissions [108].

The development of scalable and cost-effective $CO_2$ captures technologies and the widespread adoption of sustainable fuel synthesis will require the capture of large amounts of $CO_2$ from industrial processes and the atmosphere. Current $CO_2$ capture technologies are often expensive and energy-intensive, limiting their feasibility on a large scale. However, researchers are developing new capture technologies based on materials such as metal-organic frameworks and zeolites, which have the potential to significantly reduce the cost and energy requirements of $CO_2$ capture [109].

The integration of Artificial intelligence and machine learning algorithms is also expected to play a significant role in the future of $CO_2$ conversion [110]. As these algorithms become more advanced, they will be able to optimize the $CO_2$ conversion process with

greater accuracy and speed, leading to further improvements in efficiency and selectivity. These technologies are still in their early stages of development and will require significant investment in research and development to optimize their efficiency and reduce costs. Many of these technologies rely on renewable energy sources such as solar and wind power. Therefore, the increased deployment and use of renewable energy sources will be critical for the success of these technologies. These technologies must be cost-effective and scalable in order to be implemented on a large scale. The development of cost-effective systems will require a focus on reducing the costs of materials and improving the efficiency of conversion processes.

Governments and policy makers can play a critical role in supporting the deployment and adoption of these technologies through supportive policies such as incentives, subsidies, and regulations [111]. Widespread public awareness and acceptance of these technologies will be important in driving demand and accelerating their adoption.

The future of these technologies will require a combination of technological innovation, supportive policies, and public engagement in order to achieve their full potential in addressing climate change and achieving a sustainable future.

The widespread adoption of these technologies will require the development of new infrastructure to support their deployment. This includes infrastructure for carbon capture, storage, and transport, as well as new facilities for the production and distribution of sustainable fuels.

Collaboration and knowledge sharing among researchers, industry, and government entities will be crucial for the successful implementation of these technologies. This will help to foster innovation, reduce costs, and accelerate the development of new technologies and processes.

The implementation of these technologies must also take into account environmental and social considerations. For example, the development of infrastructure for carbon capture and storage must be done in a way that minimizes potential environmental impacts, and the production of sustainable fuels must be done in a way that does not compete with food production or result in other negative social impacts [112,113] Public-private partnerships can play a critical role in the development and deployment of these technologies. Governments can provide funding and supportive policies, while the industry can provide expertise, technology, and investment.

International cooperation will be necessary to address the global nature of climate change and to ensure that these technologies are deployed in a way that is equitable and sustainable for all nations.

In summary, the successful implementation of the green conversion of carbon dioxide and sustainable fuel synthesis technologies will require a combination of technological innovation [114], supportive policies, public engagement, collaboration, infrastructure development, environmental and social considerations, public-private partnerships, and international cooperation.

## 6. Conclusions

The green conversion of carbon dioxide and sustainable fuel synthesis offers a promising path toward a more sustainable and low-carbon future. By continuing to invest in research, development, and policy support, we can unlock the full potential of these technologies and accelerate the transition to a more sustainable energy system.

The production of green hydrogen, using renewable electricity sources such as wind and solar, offers a promising alternative to conventional hydrogen production methods that rely on fossil fuels. Algae-based biofuels also show potential as a sustainable alternative to conventional transportation fuels, due to their ability to sequester carbon dioxide during growth.

Furthermore, the use of carbon capture and utilization techniques, such as carbon dioxide conversion to chemicals and fuels, can help close the carbon cycle and reduce carbon emissions. The development of sustainable feedstocks and production processes,

including bioreactors and other advanced technologies, can also enhance the sustainability and carbon neutrality of fuel production.

It is crucial that policymakers, researchers, and industry stakeholders work together to support the development and implementation of these sustainable fuel production methods. This includes providing policy incentives and funding for research and development, as well as promoting the adoption of sustainable fuel production technologies.

Carbon capture by microalgae was demonstrated. In addition, certain large-scale commercial research appears promising in terms of cutting significant greenhouse gas emissions and preventing global warming. Moreover, when compared to terrestrial plants, microalgae have better carbon dioxide tolerance, carbon absorption efficiency, photosynthetic efficiency, and growth rate. Unlike traditional carbon capture systems, which are largely relevant to power plants, this strategy is appropriate for carbon emissions from the transportation system while also serving as a feedstock for biofuel production for the sector, resulting in reduced emissions.

Overall, the use of green hydrogen, algae-based biofuels, carbon capture and utilization, and sustainable production processes offer promising solutions for reducing carbon emissions and addressing the challenges of climate change. With continued investment in these areas, we can accelerate the transition to a more sustainable and low-carbon energy system.

**Author Contributions:** All authors contribute equal. All authors have read and agreed to the published version of the manuscript.

**Funding:** This research received no external funding.

**Institutional Review Board Statement:** Not applicable.

**Informed Consent Statement:** Not applicable.

**Data Availability Statement:** Not applicable.

**Conflicts of Interest:** The authors declare no conflict of interest.

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
