# Peer review of "Green Conversion of Carbon Dioxide and Sustainable Fuel Synthesis"

_fire, doi:10.3390/fire6030128_

Round 1

Reviewer 1 Report

An useful review of opportunities to advancing carbon cycle controle - that might be taken further in future work 

Author Response

Thank you very much for your time and your good comment on my submission.

Reviewer 2 Report

This is an interesting paper covering the field quite well, but I have problems with its organisation and section labelling.  And its limited conclusions 

'Carbon dioxide sequestration by microalgae' 

This is the key section  but it wanders off topic  ( from lines 184- 202) to look at all approaches, comparing electro chemical and bio systems.   It refocusses specifically on algal systems from  line 230  and in detail from line 304 onwards- your main story. 

I think you need to break up the text. 184-202 would, with some adjustment, make a good general introduction to the technology options and ways of assessing them , headed as such, and the rest would follow later as being algal CO2  conversion.  Better use of Table 1 would also be good- at present as far as I can see  it  is just cited and then left free standing 

The hydrogen section is wheeled in rather oddly, at the end.  It really ought to come before the algal section.   And the conclusion section is rather short. In the text  (lines 123- 124) you make the important claim that 

'A green fuel, unlike  fossil fuel, does not emit more CO2 into the atmosphere since its burning, in the well-to-wheel cycle, creates a CO2 release equivalent to that required for its manufacture, closing  the CO2 balance, making it a carbon-neutral fuel' 

I am not sure it is necessarily true that the energy used in CO2  conversion processes  is  always 

the same as is produced when the fuel is burnt. You says it's 'equivalent',  making it carbon neutral.  But as you point out later in the text, the various conversion processes all have different  energy  and hence carbon costs.   As far as I can see, all you can says is that  some are high and that  for algal systems they may be lower but  we don't have the data yet.  So, reasonbly enough, at the end of the alglal section  (319-320) you  say 

'To evaluate the atmospheric consequences of microalgae-based carbon se-questration, life cycle estimate schemes need to be developed'.

I think you need to round the paper off  better by commenting a bit more on these issues. At moment you just assert  that 

'when compared to terrestrial  plants, microalgae have better carbon dioxide tolerance, carbon absorption efficiency',   [line 444] 

Maybe so, but what about the other options you look including hydrogen?

Can anything more be said?  

Author Response

This is an interesting paper covering the field quite well, but I have problems with its organisation and section labelling.  And its limited conclusions 

'Carbon dioxide sequestration by microalgae' 

This is the key section  but it wanders off topic  ( from lines 184- 202) to look at all approaches, comparing electro chemical and bio systems.   It refocusses specifically on algal systems from  line 230  and in detail from line 304 onwards- your main story. 

I think you need to break up the text. 184-202 would, with some adjustment, make a good general introduction to the technology options and ways of assessing them , headed as such, and the rest would follow later as being algal CO2  conversion.  Better use of Table 1 would also be good- at present as far as I can see  it  is just cited and then left free standing 

Response: Thanks for your point of view. We have changed the text as you requested.

The hydrogen section is wheeled in rather oddly, at the end.  It really ought to come before the algal section.   And the conclusion section is rather short. In the text  (lines 123- 124) you make the important claim that 

'A green fuel, unlike  fossil fuel, does not emit more CO2 into the atmosphere since its burning, in the well-to-wheel cycle, creates a CO2 release equivalent to that required for its manufacture, closing  the CO2 balance, making it a carbon-neutral fuel' 

I am not sure it is necessarily true that the energy used in CO2  conversion processes  is  always the same as is produced when the fuel is burnt. You says it's 'equivalent',  making it carbon neutral.  But as you point out later in the text, the various conversion processes all have different  energy  and hence carbon costs.   As far as I can see, all you can says is that  some are high and that  for algal systems they may be lower but  we don't have the data yet.  So, reasonbly enough, at the end of the alglal section  (319-320) you  say 'To evaluate the atmospheric consequences of microalgae-based carbon se-questration, life cycle estimate schemes need to be developed'.

I think you need to round the paper off  better by commenting a bit more on these issues. At moment you just assert  that 'when compared to terrestrial  plants, microalgae have better carbon dioxide tolerance, carbon absorption efficiency',   [line 444] 

Response: Thanks for your point of view. We have modified the text as you requested

Maybe so, but what about the other options you look including hydrogen?

Can anything more be said?  

Response: Thanks for your point of view. We have modified the text as you requested including conclusion.